# Towards an international research agenda for public health advocacy: Practice, preparedness and knowledge gaps

Katherine Cullerton[1], Kelly D'cunha[1], Chloe Clifford Astbury[2]*, Daniel Hunt[1,3], Alexandra J. Bhatti[4], Richmond Aryeetey[5]

1 School of Public Health, The University of Queensland, Brisbane, Queensland, Australia, 2 Global Strategy Lab, York University, Toronto, Canada, 3 Centre for 21st Century Public Health, Department for Health, University of Bath, Bath, United Kingdom, 4 Department of Health Sciences, Macquarie University, Sydney, New South Wales, Australia, 5 Department of Population, Family and Reproductive Health, University of Ghana, Accra, Ghana

* castbury@yorku.ca

## Abstract

Advocacy is a core function of public health practice and is essential for advancing population health, whether by promoting preventative measures or encouraging evidence-based policy reform. However, public health advocates, globally, face considerable barriers, including limited knowledge of effective strategies, resource constraints (such as time and funding), low prioritisation within organisations, and interference from powerful commercial industries seeking to impede policy change. To better support advocacy efforts, we sought to understand global advocacy practices, identify effective strategies, and determine where additional resources or evidence are most needed. We conducted an online cross-sectional survey with 156 self-identified public health advocates across 36 countries. Most respondents (80%) reported engaging in advocacy regularly, dedicating approximately half of their work time on related activities. Working in coalitions on policy issues was one of the most frequently used and effective strategies. While 61% of respondents felt well-equipped with advocacy knowledge, many reported gaps in other areas of preparedness, including insufficient funding, lack of time, limited access to networks, and gaps in advocacy skills, particularly regarding engaging with industry. Notably, respondents from middle-income countries reported higher self-assessed advocacy skills than those from low- or high-income countries. These findings highlight the need for tailored support and resources, particularly in relation to capacity building and evidence generation. In response, we propose a research agenda to address the most pressing issues facing public health advocates, globally.

**Data availability statement:** Due to data containing potentially identifying information, the participant raw data has not been made publicly available. However, reasonable requests for access to the data can be made to the ethical committee via humanethics@research.uq.edu.au.

**Funding:** The authors received no specific funding for this work.

**Competing interests:** The authors have declared that no competing interests exist.

## Background

Translating public health research into policy and practice is a notoriously difficult task. Policy change, particularly in public health, is a complex and unpredictable process often characterised by the presence of powerful vested interests opposing reforms, and by tensions between evidence and ideological beliefs [1,2]. Public health advocates – individuals seeking to improve population health by promoting preventative measures and encouraging evidence-based policy reform, play a critical role in influencing positive change in this landscape.

Advocacy is traditionally defined as "the act of pleading for or against a cause, as well as supporting a position, a point of view or course of action" [3], which can be employed at an individual or collective level. In public health advocacy, the focus is often on collective action and driving systems change, with a clear recognition of the importance of engaging in political processes to achieve the desired policy outcomes [4].

Advocacy is recognised as a core function of public health practice and is essential for advancing population health. It is a key component of the World Health Organization's (WHO) essential public health functions to operationalise and strengthen public health systems [5]. Advocacy skills are prioritised in public health competencies such as the Association of Schools for Public Health in the European Region [6] and the Core Competencies for Public Health in Canada [7].

Effective advocacy has contributed to significant public health achievements [8]. Advocacy efforts by early HIV/AIDS activists facilitated access to affordable treatment, while advocates for the International Code on the Marketing of Breast-Milk Substitutes contributed to global protections for infant health [9]. Similarly, tobacco control advocates advanced smoke-free legislation and plain packaging policies in many countries [10]. The advocacy strategies used to achieve these outcomes were diverse, including media engagement, community mobilisation, building relationships with decision-makers, challenging corporate narratives and promoting evidence-informed policy recommendations [11].

While these positive examples should be celebrated, public health advocates continue to face significant barriers to achieving policy reform, globally. Advocacy approaches are shaped by the diversity of political environments and histories of engagement across countries. In some contexts, it is relatively safe to engage directly with government actors, whereas in others, political constraints or risks may limit such engagement. In addition to these contextual challenges, lack of resources (time and funding), low prioritisation and workforce challenges, such as insufficient training and experience, have been cited as obstacles to effective advocacy practice [12–15]. Studies indicate that public health professionals often lack the necessary skills and confidence to engage in effective advocacy efforts, which hampers their ability to address critical policy issues [15,16].

In addition, advocates often contend with the obstructive influence of powerful industries whose commercial interests are misaligned with the public interest. These industries seek to block, delay or avoid policy change that threaten their profits [17]. They do this directly, by undermining the credibility of public health advocates, and

indirectly, by influencing regulations and norms in ways that discourage government support for civil society. This reframes advocacy as an unwelcome interference in business affairs rather than a tool for promoting good governance. Keeping up with these evolving industry tactics is a constant challenge for under-resourced advocates and underscores the urgent need for a skilled public health workforce capable of engaging in effective advocacy [13].

Another impediment to the development of a skilled workforce is that advocacy practice has received limited attention both empirically and in public health curricula [12,14,18,19]. There remains limited understanding of the range of advocacy activities being undertaken internationally, which strategies are effective, and the support systems required for public health professionals to succeed as advocates [20,21]. This is partly due to challenges evaluating the diverse approaches used in advocacy and the complex interplay of factors influencing policy-making that make it difficult to attribute outcomes to a particular activity or set of actions.

To better equip the public health workforce to influence policy change, empirical research is needed to understand current advocacy practices, identify effective strategies, and determine where additional resources or evidence are required. This includes examining how advocacy is integrated into professional roles and assessing the workforce's capacity and confidence to engage in advocacy. Accordingly, this study aims to investigate global public health advocacy practices, highlight enabling conditions, and identify opportunities to strengthen advocacy capacity for improved health outcomes.

## Methods

A cross-sectional online survey was designed to collect information about public health advocates' practices and the training and research needed to support their efforts. Participants were recruited using purposive and snowball sampling. The survey was open for a 10-week period between March and June 2023.

Ethics approval for this study was received from the human research ethics committee of The University of Queensland (2022/HE001627).

### Participant recruitment

We have added a, and then by peer-nomination snowball sampling [22]. An advanced Google search was conducted to identify a sample of public health agencies working to improve health outcomes in public health priority areas such as nutrition, breastfeeding, cancer, alcohol, smoking, sexual health and mental health. Publicly available contact details for each relevant agency were collected. An invitation to participate in the survey was emailed, addressed to the CEO or manager of the organisation wherever possible. Email recipients were also invited to forward the survey link to other agencies that might be eligible and interested in participating. The survey was also promoted in newsletters of public health organisations globally and on social media.

### Inclusion and exclusion criteria

Inclusion criteria were that participants: self-identified as being involved in public health advocacy; worked for a non-government organisation (NGO), government organisation, university/research institute, United Nations (UN) agency or foundation of a for-profit organisation; and were able to comprehend and write in English. Individuals were excluded if they did not identify as a public health advocate or they worked for a private or commercial organisation. Eligible participants were included if they completed any of the survey items.

### Survey items

The survey was designed using Qualtrics software (Version 2023) with input from all authors and comprised 21 items, including multiple-choice, close-ended, and open-ended questions (see S1 File). The survey could be completed on either a desktop computer or a mobile device. Some survey response options, such as the areas of public health focus, were informed by the Public Health Advocacy Institute [23] and the American Public Health Association [24]. The first five

questions were designed to confirm the respondent's eligibility. Section two included 12 questions to collect demographic and professional data. Section three comprised four questions designed to gain a better understanding of the strategies currently used by public health advocates and identify their support needs.

To establish content validity, the survey was piloted by a panel of public health administrators and students who were not eligible to participate in this study. Panellists were asked to assess the content, usefulness and completion time of the survey using a validity assessment form. The survey was revised in response to their feedback before distribution.

### Data analysis

As the number of individuals who received the invitation could not be tracked, it was not possible to calculate a response rate. Responses provided by participants were categorised as either quantitative or qualitative data for analysis. For quantitative data, descriptive statistical analysis was performed using STATA (version 17) [25]. Frequencies for categorical data were described and summarised into counts and percentages. Content analysis was used to analyse the qualitative data. After thoroughly reading the responses, the text was coded and organised into categories, following the approach outlined by [26], with 10% of the text independently coded by two members of the research team. Disagreements were resolved through discussion or, if necessary, by a third member of the team [26,27].

### Results

Three hundred and twenty seven individuals began the survey, with 153 eligible to participate. Demographic details of respondents are summarised in Table 1. The majority of respondents identified as women (72%) and reported having worked in public health for over 10 years (63%), and primarily within NGOs (57%). Most participants (86%) reported engaging in advocacy work at a national level, while half indicated their advocacy work included a state or local level focus (participants could select more than one level of focus). The most commonly cited focus of respondents' advocacy work was chronic disease risk factors (86%), followed by environmental health (45%).

The study included participants from 36 countries (see S2 File). Australia was the most common location (n = 54), followed by Canada (n = 14). While 74% of participants were from high income-countries, the full sample represented a wide range of WHO regions (see Table 2).

While the majority of respondents identified one country as the focus of their advocacy work (often where they were located), 59% reported their work included a global or regional (e.g., Africa, Europe) focus.

### Details of advocacy in practice

Most participants (80%) reported regularly engaging in advocacy activities and often spent at least 50% of their work time on advocacy (Table 3). The most frequently reported advocacy activity in the previous 12 months was working in an advocacy coalition on a policy issue, with 63% of respondents engaging in this activity monthly or more often. Other commonly-used strategies included meeting with health civil servants (48% monthly or more) and 'other' strategies not listed, such as, social media campaigns (48%). The three strategies least frequently used were commissioned polling of public opinion (60% of participants responded 'never'), writing a letter to the editor (45% 'never') and meeting with non-health civil servants (38% 'never'). Of note, nearly one-third (30%) of participants reported they had never met with a politician about their policy issue.

### Advocacy strategies perceived as successful

Participants were asked to identify the advocacy strategy they considered most effective. Content analysis of responses found that the strategies most frequently identified as effective were building relationships with decision makers and influencers (n = 32 respondents) and forming advocacy coalitions (n = 30). Fourteen additional strategies were identified as shown in Table 4. Notably, only eight responses were coded 'a comprehensive approach with multiple strategies' as those participants stated multiple strategies in their response.

 

**Table 1.** Demographic characteristics of eligible survey respondents (n = 153).

| Characteristic | n | (%) |
|---|---|---|
| **Gender**[a] | | |
| Woman | 109 | 72% |
| Man | 38 | 26% |
| Non-binary/Non-conforming | 2 | 1% |
| Prefer not to respond | 2 | 1% |
| **Duration of work in public health** | | |
| >10 years | 96 | 63% |
| 6–10 years | 36 | 23% |
| 2–5 years | 14 | 9% |
| <2 years | 7 | 5% |
| **Sector of most advocacy work** | | |
| Non-government organisation | 87 | 57% |
| University/Research institute | 41 | 27% |
| Government organisation | 16 | 10% |
| United Nations agency | 5 | 3% |
| Other (e.g., freelance, prisons) | 4 | 3% |
| **Level of advocacy work**[b] | | |
| National | 132 | 86% |
| State/District/County | 76 | 50% |
| Local/Community-based | 71 | 46% |
| International/Global | 53 | 34% |
| Regional | 38 | 25% |
| **Focus of advocacy work**[b] | | |
| Chronic disease risk factors (e.g., food system, nutrition, physical activity, tobacco) | 131 | 86% |
| Environmental Health | 69 | 45% |
| Maternal and Child Health | 50 | 33% |
| Mental health | 43 | 28% |
| Other | 36 | 24% |
| Communicable diseases (e.g., HIV/AIDS, malaria, tuberculosis) | 32 | 21% |
| Vaccines | 27 | 18% |
| Substance Misuse | 24 | 16% |
| Injury & Violence Prevention | 21 | 14% |
| Sexual health | 18 | 12% |
| General public health and health promotion | 7 | 5% |
| Health equity (e.g., migrant and refugee health) | 5 | 3% |
| Indigenous health | 3 | 2% |

[a] n = 2, did not respond.

[b] Participants could select multiple responses if they worked at multiple levels and on different focus areas.

## Preparedness to be an effective advocate

Despite the extensive experience of many participants and their frequent engagement in advocacy activities, survey responses indicate many did not feel well equipped to be an effective advocate encompassing knowledge, skills and resources. While a majority (61%) stated they felt well equipped with advocacy knowledge, many participants reported gaps in other areas of preparedness. Specifically, 95% stated they were not well equipped or were only somewhat well

**Table 2. Countries and participants by The World Bank country classification by income level and WHO regional classification[a].**

| Classification | By country (n = 36) | | By participants (n = 153) | |
|---|---|---|---|---|
| | n | % | n | % |
| *Income levels* | | | | |
| High-income | 15 | 42 | 113 | 74 |
| Upper middle-income | 6 | 16 | 11 | 7 |
| Lower-middle income | 10 | 28 | 23 | 15 |
| Low-income | 5 | 14 | 6 | 4 |
| *WHO regions* | | | | |
| AFR | 11 | 31 | 23 | 15 |
| EUR | 7 | 19 | 25 | 16 |
| WPR | 6 | 17 | 66 | 43 |
| AMR | 6 | 17 | 27 | 18 |
| SEAR | 4 | 11 | 8 | 5 |
| EMR | 2 | 5 | 4 | 3 |

[a] AFR = African Region; EUR = European Region; WPR = Western Pacific Region; AMR = Americas; SEAR = South-East Asian Region; EMR = Eastern Mediterranean Region.

**Table 3. Frequency of use of advocacy strategies over the past 12 months.**

| Question | Regularly (Daily to Monthly), % | Irregularly (1–2 times/year) | Never |
|---|---|---|---|
| Worked in an advocacy coalition on your policy issue | 63 | 23 | 14 |
| Met with a 'health-related' civil servant/government bureaucrat regarding your policy issue | 48 | 39 | 13 |
| Other (e.g., protests, social media campaigns) | 48 | 33 | 19 |
| Met with a politician regarding your policy issue | 28 | 42 | 30 |
| Written a submission in response to a government inquiry on your policy issue | 27 | 61 | 12 |
| Met with a 'non-health' civil servant/government bureaucrat (e.g., Ministry of Finance) regarding your policy issue | 27 | 35 | 38 |
| Written a media release on your policy issue | 24 | 50 | 26 |
| Interviewed by the media on your policy issue | 21 | 55 | 24 |
| Commissioned research to inform advocacy on your policy issue | 15 | 59 | 26 |
| Written a letter to the editor on your policy issue | 12 | 43 | 45 |
| Written an article for the public on your policy issue that was published in mainstream media or online | 11 | 63 | 26 |
| Commissioned polling of public opinion on your policy issue | 4 | 36 | 60 |
| Use of any advocacy strategy[a] | 82 | 18 | 0 |
| Proportion of time spent on advocacy work engagement (mean ± standard deviation)[b] | 50 ± 30 | | |

[a] n = 130.

[b] n = 138.

**Table 4. Advocacy strategies used by participants that they considered most effective (n = 153).**

| Advocacy strategy | No. of participants identifying strategy as effective |
|---|---|
| Building relationships with decision makers and influencers | 32 |
| Forming advocacy coalitions | 30 |
| Using media/social media | 13 |
| Grassroots campaign including personal testimony from those with lived experience | 12 |
| Providing information to government officials, e.g., training, policy briefs | 10 |
| Using evidence-based facts/research including economic impact | 13 |
| Comprehensive approach with multiple strategies – including incorporating insights from theory, e.g., multiple streams theory | 8 |
| Framing/communication of message | 6 |
| Using champions/messengers including religious leaders | 4 |
| Participating in government consultations/submissions/inquiries | 4 |
| Using legislative processes | 3 |
| Presenting a range of clear, prioritised solutions | 2 |
| Utilising a crisis | 1 |
| Public opinion polling | 1 |
| Hiring a consultant with expertise | 1 |
| Joining internal/government advisory groups | 1 |

equipped with funding, 79% reported a lack of time, 64% felt they lacked access to networks, and nearly half (48%) said they were not well equipped or only somewhat well equipped with advocacy skills (Fig 1).

When considering advocacy preparedness across country levels of income (Fig 2) fewer participants from low-income countries reported being well-equipped across all aspects of advocacy compared to participants from other income groups. In particular, a greater proportion of participants from high- and middle-income countries reported themselves to be well-equipped with advocacy skills and knowledge. Notably, preparedness did not always align with national income levels. While participants from all countries reported limited access to funding, those from middle-income countries reported feeling relatively well-equipped across multiple areas (skills, knowledge, time, and funding). In contrast, participants from low-income countries indicated they felt more equipped with advocacy knowledge than in their advocacy skills, and only somewhat equipped or not at all equipped with funding and time.

Qualitative analysis of participants' comments about preparedness identified that a key reason many did not feel well equipped was that advocacy was not integrated into formal roles or supported by leadership:

*"I am a 'soft-funded' researcher, so advocacy isn't officially part of my job and takes time away from my 'real' work, so time/funding is very limited but I feel ethically obliged to speak out about the policy implications of my research findings."* (New Zealand, University/Research Institute)

*"How empowered staff feel, or actually are, based on their relationships with Medical Health Officers and other senior leadership can be an important factor. Staff can have networks, skills, knowledge, and education, but if they don't have*

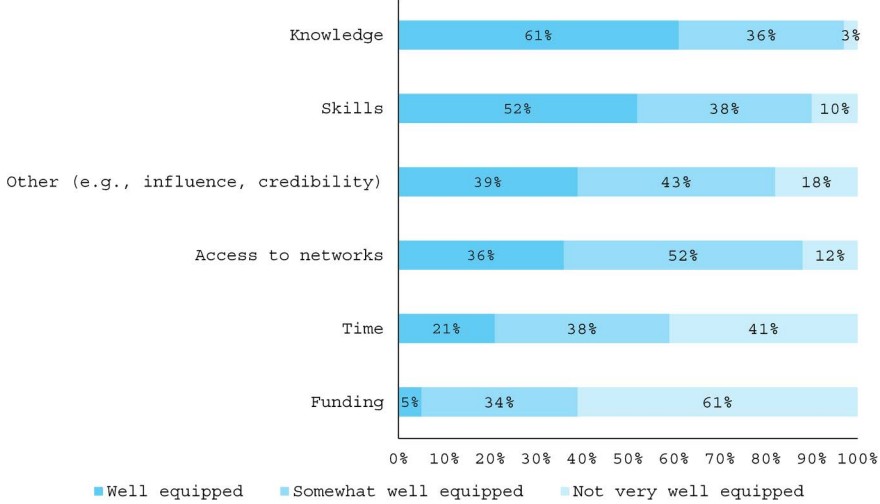

**Fig 1. Proportion of respondents by perceived preparedness to be an effective advocate.**

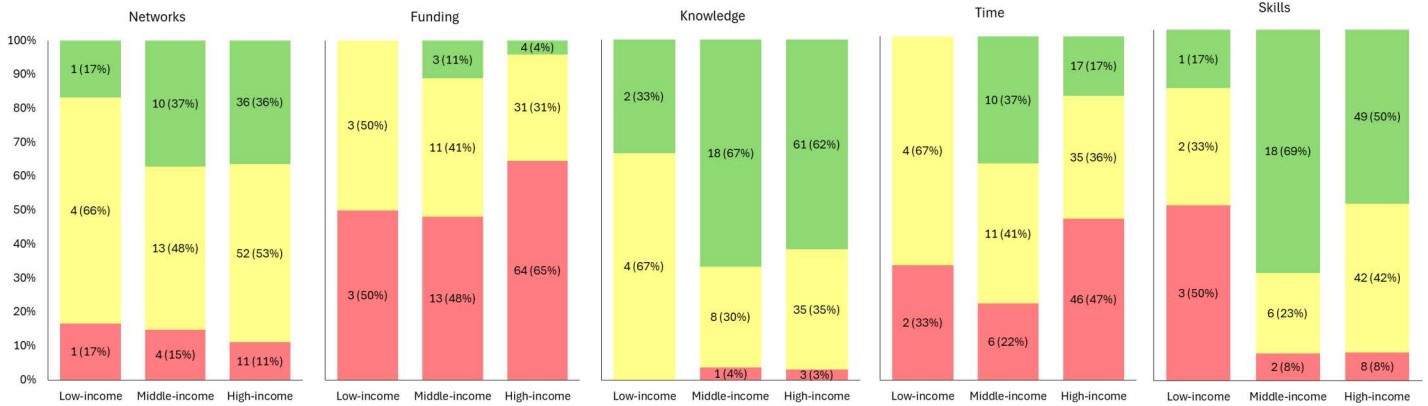

**Fig 2. Distribution (in %) of the levels of support for the statements around being 'equipped' with access to networks (n = 132), funding (n = 132), time (n = 131), knowledge (n = 132), and advocacy skills (n = 131) to be an effective advocate across World Bank levels of income.**

*the trust of senior leadership or if senior leadership doesn't really know the breadth and depth of these staff attributes, it can be squandered."* (Canada, Government organisation)

Many respondents also acknowledged the complexity of advocating in public health with resource constraints, particularly funding and time – both to engage in activities and to build the required skills.

*"We need assistance in navigating legislation and connections through the health system. Too many topics and jurisdictions to truly address issues with limited resource allocated in non-government sector."* (Australia, NGO)

*"The main barrier is lack of funding and lack of resources"* (Belgium, Government organisation)

*"It takes many years to build up a nuanced understanding of the most effective advocacy strategies and tactics"* (United Kingdom, NGO)

## Training and evidence needed to enhance advocacy practice

Half the respondents (50%) reported having undertaken formal training in developing and/or implementing effective advocacy strategies. This included training provided by a wide range of public health and advocacy institutes and universities, as well as local programs and short online courses. Key areas of training focus identified by participants included messaging, coalition building and media engagement.

Respondents identified a variety of training needs and advocacy strategies they felt needed more evidence to inform their advocacy practice. The most commonly identified needs are outlined below.

**Evaluation of advocacy strategies to identify the most impactful.** A dominant theme in participants' responses was the need for increased and more robust evaluation of advocacy action. Respondents noted this would enable them to improve their practice based on empirical evidence rather than relying on personal experience.

*Most advocacy strategies are not well researched, and results published. Grass roots advocacy and its influence on political will is known but not substantiated with good evidence.* (Australia, NGO)

*I think that knowing other experiences could greatly enrich my day-to-day work* (Mexico, UN agency)

Interestingly, one participant cautioned against promoting evidence of effective advocacy strategies, stating it would "somehow be dangerous … as nothing is stopping tobacco companies using it in their favour" (Australia, Government organisation).

**How and when to engage with government/policymakers.** Many participants expressed a need for more guidance on effectively engaging with government departments and policymakers. Some wanted to better understand mechanisms for participating in public policy-making and legislative protocols, to better determine when and how to engage with policymakers. Understanding when, and in what parts of the process, it would be most impactful to participate was particularly important for those with limited resources.

*"NGO awareness of policy process and how to influence legislation at various stages."* (Ireland, NGO)

*"Most NGOs focus on data and evidence and not enough on the policy process which is key."* (Belgium, NGO)

Participants also wanted evidence of the efficacy of efforts to engage policymakers, particularly which advocacy tools, such as written submissions for inquiries, consultations and petitions, are most effective in influencing policy change.

*"The impact of written submissions for inquiries/consultations - how do we know decision makers haven't already made up their minds."* (New Zealand, NGO)

*"I'd like to see evidence on whether signed petitions are effective at changing policy. My suspicion is that they are not, but I haven't yet seen evidence to support."* (Canada, University/Research Institute)

**Evidence of effective knowledge translation and communication.** Participants wanted more evidence on effective communication and messaging for advocacy. This included the sharing of learnings, tactics, and effective arguments and counterarguments from advocates working on similar public health issues in other contexts. Several participants also expressed interest in further research into the impact of messaging, particularly values-based messaging.

*"Advocacy strategies as it relates to countering industry messages and lobbying. Countering industry frames. How to compete with industry lobbyists who have more resources than social services."* (Canada, Government organisation)

Respondents wanted more evidence about the effectiveness of different traditional and social media strategies, including the usefulness of media releases and factors influencing media priorities and coverage.

> *"Media advocacy; I would like to know how impactful this advocacy strategy is."* (Ghana, NGO)

> *"How effective social media campaigns are in influencing policy makers."* (Kenya, NGO)

> *"Media coverage of alcohol issues is not straightforward compared to other health issues as media is not a disinterested player given their income from alcohol advertising. This is an important area to understand and to carry out systematic research."* (Ireland, NGO)

Participants also expressed a need for knowledge of effective communication tactics and messaging to increase public support for a public health policy or action.

> *"Empowering/strengthening of the community to demand accountability from duty bearers both the community level and national level."* (Ghana, NGO)

> *"Strategies to build public demand for public health policy."* (Switzerland, NGO)

**Understanding industry tactics**

Several participants emphasised the need for more knowledge of industry tactics that undermine public health. Specifically, they highlighted the importance of understanding lobbying strategies used by industry or other opponents to their policy recommendations, how these can be countered, and what health advocates can learn from industry's effective tactics.

> *"What are they doing that we could learn from – obviously they have more money but what else? How do they shut down debate?"* (New Zealand, NGO)

There were also specific requests for research investigating how commercial actors are infiltrating civil society organisations (with one citing the example of an industry lobbyist's appointment to the board of a national association of health organisations), and greater monitoring of the marketing of unhealthy food to children.

**Effective use of advocacy coalitions.** While forming or participating in advocacy coalitions was identified as a strategy regularly used by many participants in this study, several expressed a need for more research on the benefits and potential drawbacks of coalitions. Others called for a deeper exploration of the effectiveness of the actions and dynamics of coalitions.

> *"Benefits/otherwise of Coalitions: do they work better in policy windows only; how do they most effectively operate; engaging with politicians?"* (Australia, University/Research Institute)

> *"How do you create the winning team to win the fight?"* (Netherlands, NGO)

**Generating data to support advocacy.** Participants, particularly from government organisations, identified a range of data that would support their advocacy efforts. The most frequently mentioned were economic analyses to support investment in prevention:

> *"I think what has most sway with politicians is economic analysis of health care dollars spent on various forms of prevention versus the same money directed towards primary health care. Are the outcomes better? Do we reach more people? Is this good value for money?"* (Australia, Government organisation)

Other suggestions included generating local data, particularly in regions (e.g., Africa) or for population groups (e.g., Aboriginal and Torres Strait Islander peoples) that are underfunded for research. Participants also highlighted a need for case studies (local and international) of successful advocacy actions and increased policy monitoring and surveillance.

*"Funding for more in-depth research to inform policy."* (Barbados, Government organisation)

*"I need evidence of more governments who have done x and created positive impact."* (Australia, Government organisation)

*"More local evidence is required for advocacy work"* (Brunei Darussalam, Government organisation)

In addition to research aimed at building capacity, two participants emphasised the need for research exploring the pressures and challenges inherent in advocacy work. They called for researchers to be more responsive to their needs and the realities of advocacy practice:

*"Do short-term research appropriate for advocacy, i.e., not current NHMRC-type [funding body] processes where it takes a year to develop the application, a year to consider, a year to set up, etc... Advocacy timelines are much shorter than this - and advocacy research funding needs to recognise this - as do some researchers."* (Australia, University/ Research Institute)

They asked for research support that addresses practical questions and encouraged researchers to allocate time and resources to support advocates.

## Discussion

This study provides insights into the advocacy practices of public health advocates around the world and the training, support and evidence they want and need to enhance their capability and actions to enhance population health outcomes.

Participants worked in a variety of sectors, in 36 different countries. While the majority were from high-income countries, almost 30% were from low- or middle-income countries, predominantly in Africa. This diversity has provided a range of perspectives on advocacy practices, preparedness and knowledge gaps that have not been explored in previous studies of public health advocacy. Interestingly, there were few responses from advocates based in the USA despite it being the primary location for much of the existing health advocacy activity [28].

Advocates from around the world reported building relationships with decision makers and forming advocacy coalitions were the most effective advocacy strategies. However, while many invested considerable time in advocacy coalitions, and a substantial portion of participants regularly engaged with government officials or civil servants, fewer engaged with politicians. Notably, 30% of participants reported that they had never met with a politician regarding their policy issue. The strategy of investing in relationships, particularly with politicians or their staff, has been identified in other studies as a high-return investment for advocates [13,15,29,30]. It is important to note that the low engagement with politicians may reflect different interpretations of advocacy. Some participants might consider advocacy to be the influence of any decision maker, and so politicians may not be their primary target.

Activities least likely to be undertaken were commissioned polling of public opinion, writing letters to the editor, and meeting with non-health civil servants. While the first two strategies have either resource constraints or limited effectiveness with the dramatically altered media landscape [31], it was surprising to see that a significant portion (38%) of respondents 'never' engaged with non-health civil servants. This lack of cross-sectoral engagement poses a challenge for effective multisectoral and intersectoral health policymaking, particularly for health issues that fall under the responsibility of non-health portfolios, such as agriculture or urban planning. Calls for diversifying decision-maker targets have increased in recent years, not only to increase the effectiveness of advocacy [32,33] but also in line with a 'Health in All' policies approach

[34]. Encouragingly, more than a quarter of this study's participants had met with 'non-health' civil servants or bureaucrats monthly or more frequently, suggesting growing recognition of the importance of cross-sector engagement.

While a small majority of public health advocates felt confident in their advocacy knowledge, only half expressed confidence in their advocacy skills. This aligns with findings of other studies with public health professionals [12,14–16] and highlights the need for enhanced training opportunities targeting both skills and knowledge. Interestingly, respondents from middle income countries felt better equipped in their advocacy skills than those from low- or high-income countries. This finding is unexplained with no clear rationale and warrants further exploration. We found that inadequate or unsustainable funding remains a critical barrier for advocates a finding that does not appear to have been explored in other studies. Notably, advocates from high income countries in our study were more likely to report feeling under-equipped in terms of funding than their counterparts in low- and middle-income countries. While the reasons for this are unclear, we hypothesise it may reflect greater competition for funding in high-income settings, and/or differing perceptions of what it means to feel 'under-equipped'. This unexpected finding warrants further investigation.

A finding that has received less attention in the literature is the lack of prioritisation for advocacy in the workplace that many participants reported. This may be related to a lack of legitimisation for advocacy [29,35], a lack of clarity about what change can be successfully achieved, or concerns that it is too political. In another study with advocates in the UK some participants said the mandate for them to operate as political advocates was "unclear and, in some cases, uncomfortable" [15]. Given that advocacy is essential to the functioning of public health, the lack of prioritisation noted by our participants is both concerning and warrants critical attention.

Public health advocates, particularly in resource-constrained settings, emphasised their need for greater support and evidence to identify and prioritise the most impactful advocacy strategies. There was strong demand from participants for evidence of 'what works' and why, but also to build a shared understanding of advocacy among researchers and public health professionals. As noted by our respondents, there is limited empirical evidence of the effectiveness of different advocacy strategies. Drawing from the experiences of advocacy experts across diverse fields of public health as well as insights from policy mobilities, and policy process fields and political economy evaluations across different governance contexts, will be vital for developing more effective, evidence-based advocacy strategies and training opportunities that can drive meaningful change to population health outcomes.

Building the evidence of effective advocacy strategies is a critical first step. However, as our participants noted, disseminating evidence and sharing effective advocacy strategies in ways that are relevant and practical for advocates is equally crucial. This may include translating research findings into tertiary public health curricula and training academic staff to more effectively teach advocacy, including learning from advocacy practitioners and those with political expertise. Short courses and public repositories of effective strategies or mentoring are additional means of increasing advocacy skills, knowledge and confidence of the public health workforce. Adapting multiple strategies for evidence dissemination is particularly important for the heterogenous community of individuals advocating for public health.

## Limitations

Most participants in this study were working in the area of non-communicable diseases and their risk factors. The low number of participants working on communicable diseases was surprising, considering the significant global funding for this public health priority, likely driven by advocacy efforts. This is a limitation in our findings and we encourage further research to explore what insights can be drawn from this constituency for public health advocacy more broadly. Slightly fewer participants responded to the open-text questions, which may limit the depth or representativeness of qualitative insights. Additionally, due to the survey design and the requirement for anonymity, it was not possible to accurately track incomplete responses beyond eligibility screening, and we were therefore unable to provide a full attrition flow diagram.

Additionally, the survey was only available in English, and most study participants were from one country, which limited our analysis across country contexts. Consequently, we may have missed valuable insights into perceived effective

advocacy strategies as well as the role of public trust and acceptability of advocacy organisations in different government systems. We encourage future research to explore these questions across multiple languages and settings to capture a broader range of perspectives.

Finally, this study was not intended to identify causal relationships between specific strategies or combinations of strategies and outcomes; participants were asked to identify strategies with 'perceived effectiveness'. Further research is needed to measure the impact and appropriateness of these strategies in order to gain a deeper understanding of what drives successful advocacy outcomes.

## Conclusion

Advocates consistently engage in and see value in collaborating with coalitions on policy issues, highlighting the importance of collective action in advocacy efforts. To more effectively influence policy decision makers and achieve positive outcomes for population health, public health advocates need empirical evidence from quality research aligned to their needs. In this study, health advocates from around the world identified gaps that need to be addressed to build their capacity and skills to respond to current and future public health challenges. The findings suggest the following research questions may assist in mapping out future research directions:

- How do different advocacy strategies, including advocacy coalitions and communication mode and methods, influence the effectiveness of public health campaigns across varying political and social contexts?

- What are the emerging advocacy tactics used within the commercial sector, how do they influence policy and public opinion, and to what extent could these strategies be adapted to enhance public health advocacy?

- How do we safeguard and increase the legitimacy of public health advocacy in the face of evolving political, commercial and societal challenges?

- What are the most effective approaches for engaging with a variety of governments and decision-makers about public health, and how do these approaches vary based on political, cultural, and institutional contexts?

- How can data generation be optimised to support advocacy initiatives at local, national, and international levels?

- How can public health researchers and professionals develop a unified understanding of advocacy, and what strategies foster cross-disciplinary collaboration in advocating for health-promoting policies and practices?

- What are the most effective approaches to meeting the training needs of advocates, including through online platforms?

- How can advocates demonstrate their added value and effectively campaign for additional resources to support their work?

While this proposed research agenda is not designed to be comprehensive in addressing all aspects of public health advocacy, it addresses the most pressing issues identified by advocates in this study who are working to enhance population health.

## Supporting information

**S1 File. Survey questions.**
(DOCX)

**S2 File. Location of participants.**
(DOCX)

## Author contributions

**Conceptualization:** Katherine Cullerton.

**Data curation:** Katherine Cullerton, Kelly D'cunha, Chloe Clifford Astbury.

**Formal analysis:** Katherine Cullerton, Kelly D'cunha, Chloe Clifford Astbury, Alexandra J. Bhatti.

**Investigation:** Katherine Cullerton.

**Methodology:** Katherine Cullerton, Kelly D'cunha, Chloe Clifford Astbury, Daniel Hunt, Richmond Aryeetey.

**Writing – original draft:** Katherine Cullerton.

**Writing – review & editing:** Kelly D'cunha, Chloe Clifford Astbury, Daniel Hunt, Alexandra J. Bhatti, Richmond Aryeetey.

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
