## [Decision Letter · Decision Letter 0]

4 Nov 2025

PGPH-D-25-01627

Towards an international research agenda for public health advocacy: practice, preparedness and knowledge gaps

Dear Dr. Astbury,

Thank you for submitting your manuscript to PLOS Global Public Health. After careful consideration, we feel that it has merit but does not fully meet PLOS Global Public Health’s publication criteria as it currently stands. Therefore, we invite you to submit a revised version of the manuscript that addresses the points raised during the review process.

We look forward to receiving your revised manuscript.

Kind regards,

Baldeep Kaur Dhaliwal, PhD

Academic Editor

Journal Requirements:

Additional Editor Comments (if provided):

N/A

Reviewers' comments:

Reviewer's Responses to Questions

**Comments to the Author**

1. Does this manuscript meet PLOS Global Public Health’s publication criteria?

Reviewer #1: Yes

Reviewer #2: Yes

2. Has the statistical analysis been performed appropriately and rigorously?

Reviewer #1: Yes

Reviewer #2: Yes

3. Have the authors made all data underlying the findings in their manuscript fully available (please refer to the Data Availability Statement at the start of the manuscript PDF file)?

Reviewer #1: Yes

Reviewer #2: Yes

4. Is the manuscript presented in an intelligible fashion and written in standard English?

Reviewer #1: Yes

Reviewer #2: Yes

Reviewer #1: This study investigates practices, preparedness and knowledge gaps that exists in global public health advocacy through a multi-country survey and content analysis. This is a highly relevant study, which can make a significant contribution to public health advocacy literature, and holds a great value as further research can be informed through this work. The manuscript meets the publication criteria, is technically sound, and the data support the conclusions made. Appropriate data analyses have been done. Raw data is not made publicly available mentioning potential identifying information of the participants. The manuscript has a well-organized structure, and is drafted in a standard English.

However, here are some minor comments:

1. Since this is a multi-country survey, it is important to mention whether the survey form was offered in multiple languages or English only. Since participants who were able to comprehend and write in English were only included in the study, it seems like the form was developed and used in English version only. If so, restricting participation to English speakers should be acknowledged as a limitation.

2. If the survey form developed was mobile-friendly, kindly mention it in the methods section, as it clarifies accessibility issues.

3. Authors can be more transparent regarding participant recruitment and response metrics. Specifically, it does not report the number of agencies contacted, the number of individuals invited, or the overall response rate. Additionally, there is no information on attrition or incomplete responses. Including a recruitment flow diagram, similar to a CONSORT-diagram adapted for surveys, would significantly improve clarity and allow readers to better understand the sampling process and data quality.

Reviewer #2: A useful study and article. For the introduction and context, I expected to see more comment on the diversity of advocacy approaches given differing political environments and history of advocacy in countries. In some countries it is safe to advocate with the government, in some it is not. What are the implications for supporting advocacy given this diversity of settings? I think the research agenda could be more robust. What about studying on line training and support for advocates? Given that funding is a major issue, why not study how to make the case for advocacy funding? Also, given that interaction with policy makers can be highly affective and many advocates do this and many more want to, why not study what aspects/tactics with policy maker outreach is most impactful in different situations and settings? And how to communicate those findings to other advocates?

**Do you want your identity to be public for this peer review?** For information about this choice, including consent withdrawal, please see our Privacy Policy

Reviewer #1: No

Reviewer #2: **Yes:** Chris Collins

---

## [Decision Letter · Decision Letter 1]

9 Dec 2025

Towards an international research agenda for public health advocacy: practice, preparedness and knowledge gaps

PGPH-D-25-01627R1

Dear Dr. Astbury,

We are pleased to inform you that your manuscript 'Towards an international research agenda for public health advocacy: practice, preparedness and knowledge gaps' has been provisionally accepted for publication in PLOS Global Public Health.

Best regards,

Baldeep Kaur Dhaliwal, PhD

Academic Editor

Reviewer Comments (if any, and for reference):

Reviewer's Responses to Questions

**Comments to the Author**

Reviewer #2: All comments have been addressed

publication criteria?

Reviewer #2: Yes

3. Has the statistical analysis been performed appropriately and rigorously?

Reviewer #2: I don't know

4. Have the authors made all data underlying the findings in their manuscript fully available (please refer to the Data Availability Statement at the start of the manuscript PDF file)?

Reviewer #2: Yes

5. Is the manuscript presented in an intelligible fashion and written in standard English?

Reviewer #2: Yes

Reviewer #2: Thank you for addressing my comments.

**Do you want your identity to be public for this peer review?** For information about this choice, including consent withdrawal, please see our Privacy Policy

Reviewer #2: **Yes:** Chris Collins
